**Subject Category:**
Biology (whole organism)

ecology/molecular biology

population genetics, range dynamics, island biology, colonization, population dynamics

**Author for correspondence:**
Jan O. Engler
e-mail: janoliver.engler@ugent.be

†Present address: Im Mühltal 33, 61203 Reichelsheim, Germany.
‡Present address: Mägdebrink 5, 37574 Einbeck, Germany.

# Assortative mating frames establishment in a young island bird population

Jan O. Engler[1], Thomas Sacher[2,†], Timothy Coppack[2,‡] and Franz Bairlein[2]

[1]Department of Biology, Terrestrial Ecology Unit, Ghent University, 9000 Ghent, Belgium
[2]Institute of Avian Research, Vogelwarte Helgoland, 26386 Wilhelmshaven, Germany

JOE, 0000-0001-7092-1380

Successful island colonizations are key events to understand range dynamic processes, but studying a young population right after it reaches establishment is a rare opportunity in natural systems. The genetic structure of a recently established population may offer unique insights into its colonization history and demographic processes that are important for a successful colonization. Here, we studied the population genetics of a recently established island population of Eurasian blackbirds (Aves: *Turdus merula*) located on the island of Heligoland in the German North Sea. Using microsatellites, we genotyped the majority of the island population, including the nestlings, over a 4-year period between 2004 and 2007. We also genotyped high numbers of migrants on stopover and mainland individuals, as they are potential founders of the island population. We identified two genetic clusters that comply with the migrating and mainland birds. While most of the island birds belong to the mainland cluster, some breeding individuals and a low fraction of the offspring belong to the genetic cluster found in migrating individuals with almost no admixture between the two, pointing to assortative mating acting on the island population. We did not find any evidence for founder events and detected deviations from the Hardy–Weinberg equilibrium that disappeared in cohorts of older age that coincide with a lower number of siblings in older cohorts. The observed genetic patterns unravel a complex colonization history to which migratory and mainland birds have contributed and which is characterized by assortative mating. Further research will be directed towards habitat selection and phenotypic differences as potential drivers of assortative mating in this island population.

## 1. Introduction

Studying the successful colonization of remote habitats, such as islands, harbours strong potential for understanding range

dynamic processes [1], yet its genetic underpinnings are difficult to assess as examples situated at the right stage along the colonization-establishment trajectory are scarce (see [2–4] for field and experimental examples). This is why very recent colonization events may pose a high extinction risk as they have not reached establishment yet. By contrast, genetic colonization patterns from populations established a long time ago may be intertwined with local adaptation following isolation or environmental changes (e.g. [5–7], but see: [8]). Therefore, populations that just entered the establishment stage hold the strongest potential to learn about the genetic underpinnings of colonization [2–4].

By definition, the colonization process starts with the arrival of propagules (immigrating individuals) at an empty patch and ends when the extinction probability of the newly founded population no longer depends on the initial properties of the propagules [9]. From then on, a population can be considered as established. Genetically, this process is driven by the source from which propagules originate and the frequency of immigration events during the colonization stage. If propagules originate from a single source, a loss of genetic variation (i.e. a founder effect) is expected as immigrants go through a genetic bottleneck (e.g. [10,11]), meaning that not all genetic variation of the source population is also represented in the propagules. Through repeated immigration, this effect can, however, be diminished so that a newly established population does not differ genetically from the source population [12,13]. Yet, the genetics behind successful establishment can become more complex when colonization originates from more than one source, or when genetically divergent sources interbreed. In such cases, a young population could accumulate genetic variation and make it more resilient during the colonization process through admixture of beneficial genotypes (e.g. [14]). Alternatively, different genotypes might prevail without admixture during the colonization process through assortative mating which could form a starting point for sympatric speciation [15,16].

Here, we investigate the island colonization in Eurasian blackbirds (*Turdus merula*) on Heligoland Island (North Sea, Germany) that have just entered the establishment stage. Since the initial colonization in the 1980s, the population has increased in size, now comprising up to 100 breeding pairs distributed across the main island (and a maximum of 1–2 breeding pairs on the neighbouring Düne island located less than 600 m west of the main island), with breeding densities as high as in mainland populations [17,18]. Eurasian blackbirds are full migrants in Northern Europe while showing sedentary or partial migratory behaviour in Central Europe (summarized in [19], see also [20,21])—and each year thousands of individuals use the island for stopover [18]. Therefore, the island offers an interesting setting as colonizing birds could originate either from the nearby mainland or from migrating birds on stopover. Hence, our study aims to disentangle the roles of spatially (nearest mainland residents) or temporally (migrants on stopover) close individuals in the colonization success of an isolated patch of suitable habitat (i.e. the island of Heligoland).

Heligoland is located in the North Sea approximately 55 km off the German coast, making it the most isolated population of Eurasian blackbirds in Central Europe. Given the location of the island and the migration ecology of the species, we consider three colonization scenarios possible: first, the population was founded by migrating blackbirds breeding in Northern Europe which cross the North Sea during the migration and frequently stopover on the island. Second, the island was initially colonized by dispersing birds from the nearby Central European mainland populations without recruitment from stopover migrants. Finally, a combination of both scenarios could also be possible with birds from either source immigrating into the growing island population and contributing to the colonization process. To assess each of the possible colonization histories, we genotyped almost the entire island population (using microsatellites) over a 4-year period (including most of the offspring), birds that rest on Heligoland during spring migration (i.e. birds breeding in Northern Europe), and birds from close and distant mainland populations (i.e. birds breeding in Central Europe).

## 2. Methods

### 2.1. Study system

During 2004 and 2007, we colour-banded the majority of blackbirds from the island population and their offspring of which we genotyped 630 individuals (186 presumably resident adults and 444 nestlings hatched over the study period). Genotyping was conducted using seven polymorphic microsatellites (see electronic supplementary material, S1 for details on DNA extraction and genotyping). We separated birds from the presumed residents that were observed at least 20 times during weekly colour-ring counts across the study period ($n = 51$ birds, hereafter referred to as

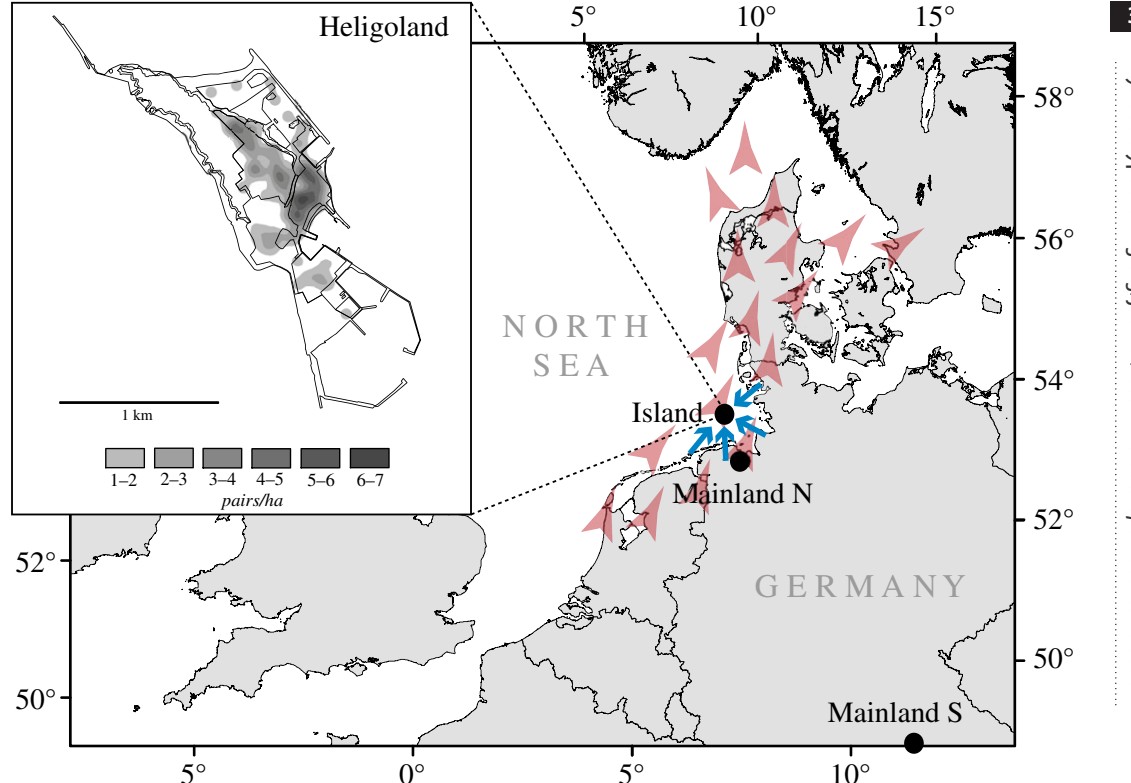

**Figure 1.** Geographical context of the study with locations of sample sites at Heligoland (Island) as well as Northern and Southern mainland locations in Germany. The two possible colonization paths of Eurasian blackbirds at Heligoland are illustrated with coloured arrows with colonization either from the nearby mainland (blue) or by migrating individuals breeding in Scandinavia (red). The inlet shows Heligoland with the shaded area representing the approximate breeding density of Eurasian blackbirds.

'residents island greater than 20'), while we aggregated the remaining 135 birds in a separate group (hereafter referred to as 'residents island less than 20'). We did this separation to (i) ensure a maximum of security in defining resident birds and (ii) to reduce negative effects of including sibling individuals in the following analyses. In addition, we genotyped 163 migrating individuals that used the island for stopover together with 29 individuals sampled from mainland Germany ($n =$ 15 from Northern Germany; $n = 14$ from Southern Germany, figure 1). While previous studies showed a weak genetic differentiation across large parts of Eastern, Central and Western Europe [19], potential genetic differences to Northern European blackbirds remain unknown. Based on Evans *et al.* [19], we expected a weak genetic differentiation between the mainland locations sampled in Northern and Southern Germany, respectively.

## 2.2. Data analysis

To convert the genotype data into the programme-specific data formats, we used TRANSFORMER T4 [22]. For each population, we then calculated the number of (effective) alleles ($N_a$ and $N_e$), observed and expected heterozygosities ($H_O$ and $H_E$) and inbreeding coefficients ($F_{IS}$) using GENALEX v. 6.502 [23,24]. We further calculated tests for deviations of group-specific Hardy–Weinberg equilibrium (HWE) and pairwise population differentiation ($F_{ST}$ and $D_{est}$) in POPGENREPORT [25]. Finally, we calculated locus-specific estimates for $H_O$, $H_E$ as well as genetic differentiation ($D$) in POPGENREPORT and supplemented these with estimates on $F_{ST}$, $F_{IS}$ using GENALEX.

## 2.3. Population assignment

To analyse the population structure and assignment, we first calculated an assignment test as implemented in GENALEX. In addition, we used STRUCTURE 2.3.3 [26,27] to infer population genetic structure of migrant, mainland and island birds without *a priori* group allocation. We estimated the most likely number

of genetic clusters ($K$) on 10 independent chains for each $K$ ranging from 1 to 10. Each chain had a length of 1 000 000 iterations where the first half was discarded as burn-in (i.e. 500 000 iterations). We assumed correlated allele frequencies and admixture, keeping the initial value of $\alpha$—the Dirichlet parameter for degree of admixture—at its default value of 1.0. To infer $K$ following the STRUCTURE runs, we used the Evanno method [28] as implemented in the online program STRUCTURE HARVESTER [29]. Initially, we ran STRUCTURE with a subset of genotyped individuals forming migrants and mainland populations as well as island birds with long record histories (i.e. 'Resident > 20' birds). This was done to avoid potential biased estimates due to deviations from HWE that were known from the full dataset (see below). We then compared these results with STRUCTURE runs using the full dataset. As the individual assignment did not change between the reduced and the full dataset for the given individuals (data not shown), we focus our results and discussion on the full dataset.

# 3. Results

Genetic diversity varied among groups with highest (effective) number of alleles in migrant birds and lowest in the southern mainland population (table 1). Overall, observed and expected heterozygosities were similar (0.708 and 0.704, respectively), although we found significant heterozygosity deficiencies for nestlings and the Southern mainland population (table 1). Deviations from HWE were found across nearly all loci for island nestlings and to a slightly lesser extent also in island birds that were observed less than 20 times (table 2). Migrant birds only showed HWE deviations in one locus ($Oe7$), while frequently observed island residents as well as the mainland populations have not shown any significant deviations from HWE (table 2). Finally, $F_{IS}$ values were always close to zero with one exception at the southern mainland location ($F_{IS} = -0.176$, table 1). Genetic differentiation among groups was consistent for $F_{ST}$ and $D_{est}$ metrics, with island groups showing lower differentiation to mainland locations than with migrants (table 3).

The population assignment test in GENALEX revealed an overall assignment to the correct population in just 42% of cases. Yet, most assignments from island groups were within the island cohort (ranging from 55.6% in residents less than 20 observations to 67.5% in nestlings), while the migration population was mostly assigned to itself (78.5% of cases, table 4). This overall picture became more resolved after the STRUCTURE analyses. By applying the delta K method as introduced by Evanno et al. [28], $K = 2$ had the lowest posterior log likelihood value paired with a high delta $K$ (electronic supplementary material, table S2). One cluster was dominated by migrant individuals, while the other cluster spreads over most (but not all) samples from the island and the close and distant mainland (figure 2). Remarkably, there was almost no admixture between the two genetic clusters at the individual level (i.e. individuals with intermediate group assignment, figure 2). In particular, the clearly assigned island offspring to either one of the two groups was unexpected under presumed random mating. Based on this finding, we ran simulations of admixture under three possible scenarios (i.e. colonization from either (i) the mainland (ii) by migrants or (iii) both sources with random mating) and calculated the unpaired mean difference from observed assignment of nestlings to assess significance (details on this approach can be found in the electronic supplementary material, S2). In brief, neither scenario resulted in clearly distinct genotypes to one of the two clusters as shown in island nestlings (electronic supplementary material, figure S1).

# 4. Discussion

Successful island colonizations offer unique insights into range dynamic processes, yet studying populations just after successful establishment are rare events in natural systems. Here, we studied the relative roles of two potential sources of propagules and found strong evidence of assortative mating that frames establishment in a young island population of Eurasian blackbirds.

While the colonization process begins with a single immigration event [9], the amount of consecutive immigration during establishment determines whether founder effects cause genetic drift in the new population or not [10–12,14]. We did not find any signs of a founder effect as observed heterozygosities were not different in island and mainland populations, while the generally higher heterozygosity found in migrant birds can be explained by the large source area (Northern Europe) to which these birds belong [18,30]. The absence of a founder effect is likely because of recurrent immigration during the establishment phase (e.g. [2,14,31]) and which is supported by the constant

**5**

**Table 1.** Population genetic parameters for genotyped individuals of each group of tested European blackbirds using seven microsatellites. $n$ refers to the number of genotyped individuals; $N_a$ refers to the number of alleles; $N_e$ refers to the expected number of alleles; $H_o$ and $H_E$ refer to the observed and expected heterozygosity, respectively (shown with the s.e.). Significant heterozygosity deficiencies are shown in italics; $F_{IS}$ refers to the inbreeding coefficient.

| genotyped blackbirds | $n$ | $N_a$ | $N_e$ | $H_o$ | ±s.e. | $H_E$ | ±s.e. | $F_{IS}$ |
|---|---|---|---|---|---|---|---|---|
| residents island (>20 obs) | 51 | 6.86 | 3.60 | 0.672 | 0.070 | 0.675 | 0.067 | 0.007 |
| residents island (<20 obs) | 135 | 9.29 | 4.08 | 0.719 | 0.030 | 0.727 | 0.042 | 0.002 |
| nestlings island | 444 | 10.71 | 3.54 | *0.650* | *0.052* | *0.680* | *0.054* | 0.044 |
| migrants | 163 | 12.14 | 6.40 | 0.795 | 0.056 | 0.806 | 0.044 | 0.020 |
| mainland north | 15 | 6.71 | 4.00 | 0.686 | 0.052 | 0.719 | 0.043 | 0.051 |
| mainland south | 14 | 4.71 | 3.15 | *0.724* | *0.075* | *0.617* | *0.062* | −0.176 |
| overall | 822 | 8.40 | 4.13 | 0.708 | 0.023 | 0.704 | 0.022 | −0.009 |

**Table 2.** Deviation from HWE for each group of birds and microsatellite locus. Significant (i.e. $p < 0.05$) deviations are shown in italic numbers.

| genotyped blackbirds | Oe1 | Oe7 | tur03 | Cu32 | LTMR6 | tur02 | tur01 |
|---|---|---|---|---|---|---|---|
| residents island (>20 obs) | 0.525 | 0.141 | 0.282 | 0.050 | 0.218 | 0.500 | 0.538 |
| residents island (<20 obs) | *0.001* | 0.420 | 0.137 | *0.001* | *0.017* | *0.009* | 0.353 |
| nestlings island | *0.000* | 0.330 | *0.003* | *0.000* | *0.000* | *0.000* | *0.007* |
| migrants | 0.899 | *0.006* | 0.969 | 0.932 | 0.142 | 0.721 | 0.142 |
| mainland north | 0.539 | 0.320 | 0.482 | 0.269 | 0.126 | 0.235 | 0.330 |
| mainland south | 0.980 | 0.736 | 0.567 | 0.945 | 0.294 | 0.094 | 1.000 |

population growth that started with only a single breeding pair [18,32]. For the same reason, we can exclude the alternative explanation that the initial colonization occurred with a large number of putative propagules (e.g. [5]).

Once a population reaches its carrying capacity, later attempts of immigration could then be blocked either stochastically by density-dependent priority effects (density blocking hypothesis, [33]) or functionally, when immigrants are less adapted than their local counterparts (immigrant inviability hypothesis, [34]). Yet, both hypotheses do not take microgeographic variation of the colonizing environment into account, which could support different locally adapted genotypes even at small spatial scales. Examples of island endemic birds illustrate that either local adaptation [35] or secondary contact [36] can drive intraspecific divergence—even at such microgeographic scales. Our results strongly point towards intraspecific divergence, driven by assortative mating as most offspring were homozygotic to one of the two clusters and which contradicted our expectations under random mating as tested through our simulations (electronic supplementary material, S2). This result is quite surprising as we expected a very low genotypic [19] and phenotypic [20] variability of blackbirds in our study region. On the other hand, Eurasian blackbirds are an iconic example of avian urbanization that accompanied notable changes in behaviour, breeding phenology and migration (e.g. [19,37,38]). Heligoland is strongly urbanized and the urban area harbours most of the blackbird population [17], yet some breeding pairs were located in rather secondary coastal shrub habitat with low anthropogenic pressure. Hence, while the exact reasons driving assortative mating in blackbirds have not been identified yet, divergent microgeographic habitat preferences could force intraspecific divergence in Heligolandian blackbirds. To this end, our next steps would be to quantify local habitat conditions at the breeding sites into individuals belonging to either genotype (i.e. intraspecific niche differentiation [39]) and possible phenotypic and phenological differences (sexual imprinting theory [40]) in this island population.

In conclusion, the observed patterns found in the breeding population of Eurasian blackbirds on Heligoland indicate a complex colonization history during which effects of genetic drift were suppressed, likely due to consecutive immigration in the early stages of establishment. At which point the second cluster arrived on the island but how persistent this small fraction of birds belonging to this genotype will be on the island is, however, difficult to ascertain. Yet the clear distinction of both clusters found in (putatively) resident birds and their offspring through assortative mating offers interesting insights into the role of different source populations in the subsequent evolution of young but isolated populations. While the genetic structure found in the island population point to predominant reproductive barriers, occasional interbreeding cannot be ruled out when population dynamics or environmental conditions change. Under such conditions, one genotype might act as a nearby evolutionary backup for the other through admixture. Hence, multiple locally adapted propagules of the same species that successfully colonized an island could make an isolated population as a whole more resilient to changing conditions, while these propagules otherwise remain isolated when conditions are stable. To this end, the detected differentiation found in Heligolandian blackbirds might vanish at later stages in the population history, when reproductive isolation disappears in the wake of environmental or habitat changes (as shown for many sympatric species; see [41] and references therein). Such an altering event might not be detectable any more afterwards, or may be hard to explain retrospectively after populations have merged. Future research needs to focus on the reasons for assortative mating in Heligolandian blackbirds and whether it correlates with

**Table 3.** Pairwise genetic differentiation among groups of European blackbird populations using $F_{ST}$ (lower triangle) and $D_{est}$ (upper triangle). The colour code stretches from blue (low differentiation) to red (high differentiation). Broader grouping is labelled in the diagonal between both triangles (in italics).

| $D_{est}$ | residents island (>20 obs) | residents island (<20 obs) | nestlings island | migrants | mainland north | mainland south |
|---|---|---|---|---|---|---|
| **$F_{ST}$** | | | | | | |
| residents island (>20 obs) | *island* | 0.018 | 0.008 | 0.248 | 0.071 | 0.047 |
| residents island (<20 obs) | 0.009 | *island* | 0.020 | 0.177 | 0.059 | 0.061 |
| nestlings island | 0.005 | 0.006 | *island* | 0.276 | 0.068 | 0.033 |
| migrants | 0.051 | 0.029 | 0.050 | *N. Europe* | 0.167 | 0.331 |
| mainland north | 0.028 | 0.019 | 0.023 | 0.032 | *C. Europe* | 0.121 |
| mainland south | 0.024 | 0.024 | 0.017 | 0.071 | 0.042 | *C. Europe* |

**Table 4.** Population assignment test. Shaded boxes refer to island bird groups (upper left) and to Central European mainland birds (lower right). Italicized values refer to same-population assignments.

| assigned populations | original populations | | | | | |
|---|---|---|---|---|---|---|
| | residents island (>20 obs) (%) | residents island (<20 obs) (%) | nestlings island (%) | migrants (%) | mainland north (%) | mainland south (%) |
| residents island (>20 obs) | *37.25* | 18.52 | 18.24 | 3.68 | 20.00 | 7.14 |
| residents island (<20 obs) | 13.73 | *19.26* | 13.51 | 7.98 | 6.67 | 7.14 |
| nestlings island | 15.69 | 17.78 | *35.81* | 3.68 | 26.67 | 21.43 |
| migrants | 9.80 | 20.74 | 3.60 | *78.53* | 13.33 | 0.00 |
| mainland north | 1.96 | 2.22 | 9.46 | 3.68 | *33.33* | 0.00 |
| mainland south | 21.57 | 21.48 | 19.37 | 2.45 | 0.00 | *64.29* |
| grand total | 100.00 | 100.00 | 100.00 | 100.00 | 100.00 | 100.00 |

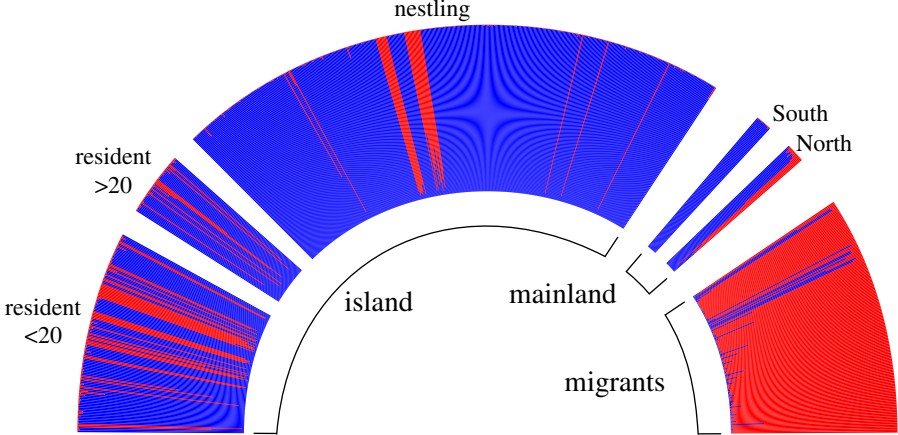

**Figure 2.** Structure barplot of all 630 genotyped individual blackbirds with $K = 2$. Among island individuals, most individuals assigned to one genotype (blue). Reference samples from two mainland populations (North and South) also merely assigned to the blue cluster, whereas in migrants, most individuals clustered to the second genotype (red).

differences in habitat selection or phenotypes (e.g. morphology or song, see [42]) to reach more functional insights of such an ecoevolutionary resilience system in island colonization.

Ethics. All research meets the ethical guidelines and legal requirements of the study country. Permission for capture of blackbirds and DNA collection was granted through the Federal Ministry of Environment, Nature Conservation, and Agriculture, Schleswig-Holstein, Kiel, Germany (permit no: V 742-72241.123-11).

Data accessibility. The genotype data are available through the Dryad Digital Repository: https://doi.org/10.5061/dryad.k24b5k2 [43].

Authors' contributions. F.B. and T.C. conceived and designed the study with input from T.S. and J.O.E. T.S. and T.C. collected field data with help from J.O.E. T.S. carried out the molecular laboratory work and allele scoring. J.O.E. carried out the data analysis and wrote the manuscript. All authors contributed to subsequent drafts and gave final approval for publication. F.B. and T.C. coordinated and secured project funding.

Competing interests. The authors declare that they have no conflict of interest.

Funding. The project was funded by the German Science Foundation (DFG; BA816/17-1). J.O.E. received additional funds by a Postdoctoral Fellowship from the Research Foundation—Flanders (FWO; 12G4317N). T.C. acknowledges additional support received through a postdoctoral grant from the Max Planck Society.

Acknowledgements. We are indebted to all ornithologists and volunteers on Heligoland who contributed to the collection of ring recoveries and observational data over the years. We are pleased to Lukas Keller and his team for supporting initial data preparation. We thank Gernot Segelbacher, the TEREC paperclub, as well as two anonymous reviewers for critical feedback on an earlier version of this manuscript.

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
