## [Reviewer comments · Royal Society Open Science]

Review History

RSOS-190050.R0 (Original submission)

Review form: Reviewer 1

Is the manuscript scientifically sound in its present form?

No

Are the interpretations and conclusions justified by the results?

No

Is the language acceptable?

Yes

Is it clear how to access all supporting data?

No

Do you have any ethical concerns with this paper?

No

Have you any concerns about statistical analyses in this paper?

Yes

Recommendation?

Major revision is needed (please make suggestions in comments)

Comments to the Author(s)

Dear editor,

I have reviewed the paper "Assortative mating frames establishment in a young island bird population." The manuscript is well written and the methods are nicely explained. I am also impressed by the amount of work; genotyping no less than 630 individuals. However, I do have some comments and suggestions regarding the methods applied to analyze the genetic data.

First, the STRUCTURE analysis is based on all loci and all individuals. Using this complete dataset can potentially bias the results. Several loci deviate from Hardy-Weinberg equilibrium while STRUCTURE assumes that loci within populations are in Hardy-Weinberg equilibrium (see introduction of STRUCTURE manual). Next, the analysis of related individuals can cause estimation biases because of shared variation that consequently affects the ancestry analysis (see for example Porras-Hurtado et al. 2013 *Frontiers in Genetics*). Given that the authors have sampled almost the entire island population; the dataset likely contains several related individuals. I would suggest removing these individuals and rerunning the analyses. You could remove related individuals with ML-related (Kalinowski et al. 2006 *Molecular Ecology Notes*). Removing related individuals might also solve the issues with Hardy-Weinberg equilibrium. Alternatively, the analyses can be done with the software package CLUSTER_DIST which does not make Hardy-Weinberg assumptions and can deal with related individuals (Rodriguez-Ramilo et al. 2014 *Genetics Research*).

During the quantification of population differentiation (using F_{st} and D_{est}), the samples are divided over six groups. Why did you not pool all the island samples (resident island [>20 obs], resident island [<20 obs] and nestlings)?

Based on the observed heterozygosity values, the authors conclude that there are no signs of a founder effect. Although I agree with this reasoning, this statement can be tested statistically. One could for example use the software BOTTLENECK, which checks if there has been a significant decrease in allelic richness (Cornuet & Luikart 1997 *Genetics*).

The supplementary material contains simulations of admixture to test particular colonization scenarios. This analyses culminated in the following sentence: "neither scenario resulted in a comparable pattern than the clearly distinct distribution of genotypes in island offspring observed." Hence, I do not see the point of including these simulations in the manuscript. Normally, one would simulate several scenarios and compare the output with the actual data using a goodness-of-fit test. The scenario that most closely matches the data gives some insights into the possible history of the population. I would suggest that either the authors remove the simulations from the paper or they perform a more detailed analyses with statistical validation.

Specific comments

Line 18: replace at with on.

Line 18: Throughout the manuscript, blackbirds is sometimes written with capital letter and sometimes with small letter. Please check for consistency.

Line 19: What genetic markers did you use? Mention the microsatellites here.

Line 21-23: Restructure this sentence. I would suggest: We also genotyped high numbers of

migrants on stopover and nearby mainland individuals, as they are potential founders of the island population.

Line 27: typo – found should be find

Line 28: What do you mean with vanished? I would use another word here or rephrase it.

Line 42: replace by with with.

Line 52: replace run with go

Line 53: replace the entire with all

Line 57-58: What do you mean with “divergent entities intervene”?

Line 65: replace holding with comprising

Line 70-71: Rephrase. I would suggest: In this regard, the system offers an interesting setting as the possible source for colonizing the island are birds from ...”

Line 76: This island population is the most isolated population of Eurasian blackbirds in Central Europe. Do you have a reference for that?

Line 84: Mention that you used microsatellites here.

Line 103-105: Move this sentence to the beginning of the section. It’s important to know how many markers you used from the beginning.

Line 104: typo – polymorphic

Line 143: It is not clear from the text how you performed the population assignment test. I guess you compared the STRUCTURE output with the origin of the samples.

Line 175-177: Can you really distinguish between recurrent migration and several propagules that intermixed later on?

Line 186: replace was with were

Line 187: I would not use the word “genotype” here. Cluster would be a better term (see also Line 204 and 206).

Line 188-190: How do your measures of genetic divergence relate to other studies on island populations? It would be interesting to compare your F_{st} -values with previous studies.

Line 195: replace elaborated with identified

Line 212-213: Because you are dealing with one species, admixture might be a better term than hybridization (which implies different species or subspecies).

Line 216-219: This sentence is difficult to follow. Please rephrase.

Review form: Reviewer 2

Is the manuscript scientifically sound in its present form?

Yes

Are the interpretations and conclusions justified by the results?

Yes

Is the language acceptable?

Yes

Is it clear how to access all supporting data?

Yes

Do you have any ethical concerns with this paper?

No

Have you any concerns about statistical analyses in this paper?

No

Recommendation?

Accept with minor revision (please list in comments)

Comments to the Author(s)

This is a very clear paper. I would only like to see more named examples elaborated on in Discussion (and Introduction) apart from one *Zosterops* and Darwin's finches.

minor issues include

23 clusters

27 did not find

28 (and more often): deviations from the Hardy-Weinberg equilibrium

32 Which phenotypic traits namely? Song?

48 What do you mean by "initial state"?

52 Such genetic bottleneck is also called founder effect. Please add this term.

64 Heligoland is an archipelago, not a single island. Specify at least here, if blackbirds settled/were sampled on the main island only.

109 (and more often in text and tables): Make sure to use subscripts and appropriate case in population genetic parameters.

123 from one to ten

139 deviation

Decision letter (RSOS-190050.R0)

12-Mar-2019

Dear Dr Engler,

The editors assigned to your paper ("Assortative mating frames establishment in a young island bird population") have now received comments from reviewers. We would like you to revise your paper in accordance with the referee and Associate Editor suggestions which can be found below (not including confidential reports to the Editor). Please note this decision does not guarantee eventual acceptance.

Please submit a copy of your revised paper before 04-Apr-2019. Please note that the revision deadline will expire at 00.00am on this date. If we do not hear from you within this time then it will be assumed that the paper has been withdrawn. In exceptional circumstances, extensions may be possible if agreed with the Editorial Office in advance. We do not allow multiple rounds of revision so we urge you to make every effort to fully address all of the comments at this stage. If deemed necessary by the Editors, your manuscript will be sent back to one or more of the original reviewers for assessment. If the original reviewers are not available, we may invite new reviewers.

When submitting your revised manuscript, you must respond to the comments made by the

referees and upload a file "Response to Referees" in "Section 6 - File Upload". Please use this to document how you have responded to the comments, and the adjustments you have made. In order to expedite the processing of the revised manuscript, please be as specific as possible in your response.

- Data accessibility

If you wish to submit your supporting data or code to Dryad (<http://datadryad.org/>), or modify your current submission to dryad, please use the following link:
<http://datadryad.org/submit?journalID=RSOS&manu=RSOS-190050>

- Competing interests

- Authors' contributions

- Acknowledgements

- Funding statement

on behalf of Professor Michael Bruford (Associate Editor) and Kevin Padian (Subject Editor)
openscience@royalsociety.org

Associate Editor's comments (Professor Michael Bruford):

Your paper has been reviewed by two referees who see strong merit in the study, but Referee 1 suggests some re-analysis, which seem sensible to me. I therefore am recommending that you at least try these analyses to see if they change the results and report this in your revision.

Comments to Author:

Reviewers' Comments to Author:

Reviewer: 1

Comments to the Author(s)

Dear editor,

I have reviewed the paper "Assortative mating frames establishment in a young island bird population." The manuscript is well written and the methods are nicely explained. I am also impressed by the amount of work; genotyping no less than 630 individuals. However, I do have some comments and suggestions regarding the methods applied to analyze the genetic data.

First, the STRUCTURE analysis is based on all loci and all individuals. Using this complete dataset can potentially bias the results. Several loci deviate from Hardy-Weinberg equilibrium while STRUCTURE assumes that loci within populations are in Hardy-Weinberg equilibrium (see introduction of STRUCTURE manual). Next, the analysis of related individuals can cause estimation biases because of shared variation that consequently affects the ancestry analysis (see for example Porras-Hurtado et al. 2013 *Frontiers in Genetics*). Given that the authors have sampled almost the entire island population; the dataset likely contains several related individuals. I would suggest removing these individuals and rerunning the analyses. You could remove related individuals with ML-related (Kalinowski et al. 2006 *Molecular Ecology Notes*). Removing related individuals might also solve the issues with Hardy-Weinberg equilibrium. Alternatively, the analyses can be done with the software package CLUSTER_DIST which does not make Hardy-Weinberg assumptions and can deal with related individuals (Rodriguez-Ramilo et al. 2014 *Genetics Research*).

During the quantification of population differentiation (using F_{st} and D_{st}), the samples are

divided over six groups. Why did you not pool all the island samples (resident island [>20 obs], resident island [<20 obs] and nestlings)?

Based on the observed heterozygosity values, the authors conclude that there are no signs of a founder effect. Although I agree with this reasoning, this statement can be tested statistically. One could for example use the software BOTTLENECK, which checks if there has been a significant decrease in allelic richness (Cornuet & Luikart 1997 Genetics).

The supplementary material contains simulations of admixture to test particular colonization scenarios. This analyses culminated in the following sentence: "neither scenario resulted in a comparable pattern than the clearly distinct distribution of genotypes in island offspring observed." Hence, I do not see the point of including these simulations in the manuscript. Normally, one would simulate several scenarios and compare the output with the actual data using a goodness-of-fit test. The scenario that most closely matches the data gives some insights into the possible history of the population. I would suggest that either the authors remove the simulations from the paper or they perform a more detailed analyses with statistical validation.

Specific comments

Line 18: replace at with on.

Line 18: Throughout the manuscript, blackbirds is sometimes written with capital letter and sometimes with small letter. Please check for consistency.

Line 19: What genetic markers did you use? Mention the microsatellites here.

Line 21-23: Restructure this sentence. I would suggest: We also genotyped high numbers of migrants on stopover and nearby mainland individuals, as they are potential founders of the island population.

Line 27: typo - found should be find

Line 28: What do you mean with vanished? I would use another word here or rephrase it.

Line 42: replace by with with.

Line 52: replace run with go

Line 53: replace the entire with all

Line 57-58: What do you mean with "divergent entities intervene"?

Line 65: replace holding with comprising

Line 70-71: Rephrase. I would suggest: In this regard, the system offers an interesting setting as the possible source for colonizing the island are birds from ..."

Line 76: This island population is the most isolated population of Eurasian blackbirds in Central Europe. Do you have a reference for that?

Line 84: Mention that you used microsatellites here.

Line 103-105: Move this sentence to the beginning of the section. It's important to know how many markers you used from the beginning.

Line 104: typo - polymorphic

Line 143: It is not clear from the text how you performed the population assignment test. I guess you compared the STRUCTURE output with the origin of the samples.

Line 175-177: Can you really distinguish between recurrent migration and several propagules that intermixed later on?

Line 186: replace was with were

Line 187: I would not use the word "genotype" here. Cluster would be a better term (see also Line 204 and 206).

Line 188-190: How do your measures of genetic divergence relate to other studies on island populations? It would be interesting to compare your Fst-values with previous studies.

Line 195: replace elaborated with identified

Line 212-213: Because you are dealing with one species, admixture might be a better term than hybridization (which implies different species or subspecies).

Line 216-219: This sentence is difficult to follow. Please rephrase.

Reviewer: 2

Comments to the Author(s)

This is a very clear paper. I would only like to see more named examples elaborated on in Discussion (and Introduction) apart from one Zosterops and Darwin's finches.

minor issues include

23 clusters

27 did not find

28 (and more often): deviations from the Hardy-Weinberg equilibrium

32 Which phenotypic traits namely? Song?

48 What do you mean by "initial state"?

52 Such genetic bottleneck is also called founder effect. Please add this term.

64 Heligoland is an archipelago, not a single island. Specify at least here, if blackbirds settled/were sampled on the main island only.

109 (and more often in text and tables): Make sure to use subscripts and appropriate case in population genetic parameters.

123 from one to ten

139 deviation

Author's Response to Decision Letter for (RSOS-190050.R0)

See Appendix A.

RSOS-190050.R1 (Revision)

Review form: Reviewer 1

Is the manuscript scientifically sound in its present form?

Yes

Are the interpretations and conclusions justified by the results?

Yes

Is the language acceptable?

Yes

Is it clear how to access all supporting data?

Yes

Do you have any ethical concerns with this paper?

No

Have you any concerns about statistical analyses in this paper?

No

Recommendation?

Accept with minor revision (please list in comments)

Comments to the Author(s)

The authors have nicely addressed all my concerns. They performed extra analyses to show that the deviations from Hardy-Weinberg equilibrium did not affect the STRUCTURE analyses. And they added a statistical procedure to the simulations. Although I would have liked a statistical test of a potential bottleneck, I understand that this was not feasible with the present data set. I think this manuscript is almost ready for publication. I did, however, find a few minor mistakes in the text (see below). Notably, the results section switches between past and present tense. This can easily be corrected.

Minor comments

Line 76: remove from

Line 109: Individuals should be lower-case

Line 138: Structure should be STRUCTURE

Line 139: replace run with ran

Line 163: replace where with were

Line 219: replace of with into

Line 243: replace united with merged

Line 243: replace need with needs

Decision letter (RSOS-190050.R1)

13-Jun-2019

Dear Dr Engler:

On behalf of the Editors, I am pleased to inform you that your Manuscript RSOS-190050.R1 entitled "Assortative mating frames establishment in a young island bird population" has been accepted for publication in Royal Society Open Science subject to minor revision in accordance with the referee suggestions. Please find the referees' comments at the end of this email.

The reviewers and Subject Editor have recommended publication, but also suggest some minor revisions to your manuscript. Therefore, I invite you to respond to the comments and revise your manuscript.

- Ethics statement

- Data accessibility

It is a condition of publication that all supporting data are made available either as supplementary information or preferably in a suitable permanent repository. The data accessibility section should state where the article's supporting data can be accessed. This section should also include details, where possible of where to access other relevant research materials such as statistical tools, protocols, software etc can be accessed. If the data has been deposited in an external repository this section should list the database, accession number and link to the DOI for all data from the article that has been made publicly available. Data sets that have been

deposited in an external repository and have a DOI should also be appropriately cited in the manuscript and included in the reference list.

If you wish to submit your supporting data or code to Dryad (<http://datadryad.org/>), or modify your current submission to dryad, please use the following link:
<http://datadryad.org/submit?journalID=RSOS&manu=RSOS-190050.R1>

- **Competing interests**

- **Authors' contributions**

- **Acknowledgements**

- **Funding statement**

Because the schedule for publication is very tight, it is a condition of publication that you submit the revised version of your manuscript before 22-Jun-2019. Please note that the revision deadline will expire at 00.00am on this date. If you do not think you will be able to meet this date please let me know immediately.

When submitting your revised manuscript, you will be able to respond to the comments made by the referees and upload a file "Response to Referees" in "Section 6 - File Upload". You can use this

to document any changes you make to the original manuscript. In order to expedite the processing of the revised manuscript, please be as specific as possible in your response to the referees.

on behalf of Professor Michael Bruford (Associate Editor) and Kevin Padian (Subject Editor)
openscience@royalsociety.org

Reviewer comments to Author:
Reviewer: 1

Comments to the Author(s)

The authors have nicely addressed all my concerns. They performed extra analyses to show that the deviations from Hardy-Weinberg equilibrium did not affect the STRUCTURE analyses. And they added a statistical procedure to the simulations. Although I would have liked a statistical test of a potential bottleneck, I understand that this was not feasible with the present data set. I think this manuscript is almost ready for publication. I did, however, find a few minor mistakes in the text (see below). Notably, the results section switches between past and present tense. This can easily be corrected.

Minor comments

Line 76: remove from

Line 109: Individuals should be lower-case

Line 138: Structure should be STRUCTURE

Line 139: replace run with ran

Line 163: replace where with were

Line 219: replace of with into

Line 243: replace united with merged

Line 243: replace need with needs

Author's Response to Decision Letter for (RSOS-190050.R1)

See Appendix B.

Decision letter (RSOS-190050.R2)

17-Jul-2019

Dear Dr Engler,

I am pleased to inform you that your manuscript entitled "Assortative mating frames establishment in a young island bird population" is now accepted for publication in Royal Society Open Science.

Kind regards,

Alice Power

Editorial Coordinator

on behalf of Professor Michael Bruford (Associate Editor) and Kevin Padian (Subject Editor)
openscience@royalsociety.org

Appendix A

Associate Editor's comments (Professor Michael Bruford):

Your paper has been reviewed by two referees who see strong merit in the study, but Referee 1 suggests some re-analysis, which seem sensible to me. I therefore am recommending that you at least try these analyses to see if they change the results and report this in your revision.

Dear Editor,

We have now prepared a revision based on the reviewers feedback. To ease their job of accessing our changes you will find the revision in track change mode. In the following you will find a detailed response to all issues raised. We are confident that our revision accounted for all comments in a detailed and appropriate way and looking forward to your decision.

Kind regards,

Jan Engler et al.

Comments to Author:

Reviewers' Comments to Author:

Reviewer: 1

Comments to the Author(s)

Dear editor,

I have reviewed the paper "Assortative mating frames establishment in a young island bird population." The manuscript is well written and the methods are nicely explained. I am also impressed by the amount of work; genotyping no less than 630 individuals. However, I do have some comments and suggestions regarding the methods applied to analyze the genetic data. We would like to thank the reviewer in providing a highly valuable and constructive review on our work. We think this critical evaluation substantially improved the work. We provide detailed responses following each paragraph.

First, the STRUCTURE analysis is based on all loci and all individuals. Using this complete dataset can potentially bias the results. Several loci deviate from Hardy-Weinberg equilibrium while STRUCTURE assumes that loci within populations are in Hardy-Weinberg equilibrium (see introduction of STRUCTURE manual). Next, the analysis of related individuals can cause estimation biases because of shared variation that consequently affects the ancestry analysis (see for example Porras-Hurtado et al. 2013 Frontiers in Genetics). Given that the authors have sampled almost the entire island population; the dataset likely contains several related individuals. I would suggest removing these individuals and rerunning the analyses. You could remove related individuals with ML-related (Kalinowski et al. 2006 Molecular Ecology Notes). Removing related individuals might also solve the issues with Hardy-Weinberg equilibrium. Alternatively, the analyses can be done with the software package CLUSTER_DIST which does not make Hardy-Weinberg assumptions and can deal with related individuals (Rodriguez-Ramilo et al. 2014 Genetics Research).

We thank the reviewer pointing to this crucial aspect. We were aware of the potential bias included by very related individuals. This was partly the reason why we separated individuals with a high record history (i.e. the "residents island >20") with those of low record history and nestlings. Regarding Structure, we - however - haven't done a separate run including only individuals from groups showing no HWE deviations (namely migrants,

mainland birds and resident >20 individuals). We did now. As you will see in the figure below, the changes from the reduced dataset with the full dataset are only marginal and strongest for the very few individuals with very insecure assignments.

Therefore, we are confident that the full dataset - while including siblings - will not affect results in a dramatic way so that the conclusions drawn remain unaffected. We now added more information why splitting the island residents into subgroups and also added our reduced structure analysis for full transparency to the reader. While we certainly could add this figure also to the appendix, we decided against because we deem it not necessary as it does not contribute to the main study question. We keep the reporting of all resident subgroups including all nestlings, as we wanted to know the assignment of each individual and hence the fraction to which these clusters are represented at each population level.

During the quantification of population differentiation (using F_{st} and $Dest$), the samples are divided over six groups. Why did you not pool all the island samples (resident island [>20 obs], resident island [<20 obs] and nestlings)?

As we already explained in the method section (which is now expanded and clarified), we wanted to ensure that we did not wrongly assign birds as "resident" what were color banded but left the island (i.e. migrant or vagrant) or regularly return to it for unknown reasons apart from reproduction. Also this separation ensured the reduction of closely related individuals as shown in the HWE results in Table 2. So we applied this division to all analyses. Hence, comparing the population differentiation (and assignment) should be safest when using the "residents island >20 " birds. Indeed, if comparing the general outcomes of the three different cohorts of island birds with migrants or mainland populations the general conclusions remain the same (i.e. island bird differ more from migrants than mainland populations). Yet, there is slight variation between residents with <20 observations to nestlings and those with >20 observations in both differentiation and assignment to other populations which is worth further investigation. Pooling all samples (for all the reasons explained above) could affect the general outcome or at least add

more variation we can better handle (and explain) using the separated cohorts.

Based on the observed heterozygosity values, the authors conclude that there are no signs of a founder effect. Although I agree with this reasoning, this statement can be tested statistically. One could for example use the software BOTTLENECK, which checks if there has been a significant decrease in allelic richness (Cornuet & Luikart 1997 Genetics). We initially thought of using Bottleneck or an M-ratio test to statistically prove the presence/absence of a founder effect. However, the utility of these methods is restricted and we see a high chance of erroneous results using them due to a lack of power in both the number of individuals (focusing on unrelated island bird) and microsatellites used here (see Peery et al. 2012 and Hoban et al. 2013 for an extensive discussion). Because of this, we decided to restrict the investigation of a potential founder effect to its discussion based on the observed heterozygosities and refrain from explicit (and testable) hypotheses of this topic.

The supplementary material contains simulations of admixture to test particular colonization scenarios. This analyses culminated in the following sentence: "neither scenario resulted in a comparable pattern than the clearly distinct distribution of genotypes in island offspring observed." Hence, I do not see the point of including these simulations in the manuscript. Normally, one would simulate several scenarios and compare the output with the actual data using a goodness-of-fit test. The scenario that most closely matches the data gives some insights into the possible history of the population. I would suggest that either the authors remove the simulations from the paper or they perform a more detailed analyses with statistical validation.

We agree that we should have added a statistical procedure to quantify the outcome of the simulation apart from a descriptive explanation. Hence, we added a comparison based on the unpaired mean difference of assignment probabilities between the island nestlings and each scenario. For each comparison a bootstrap confidence interval based on 5000 iterations is calculated. For the revision, we also redraw Figure S1 in order to better illustrate the differences and their significance. The simulation is of high importance to proof that the absence of admixture in island nestlings is not a matter of random mating of existing genotypes stemming from different sources.

Specific comments

Line 18: replace at with on.
changed

Line 18: Throughout the manuscript, blackbirds is sometimes written with capital letter and sometimes with small letter. Please check for consistency.
unified

Line 19: What genetic markers did you use? Mention the microsatellites here.
changed

Line 21-23: Restructure this sentence. I would suggest: We also genotyped high numbers of migrants on stopover and nearby mainland individuals, as they are potential founders of the island population.
Thank you, we used your suggestion.

Line 27: typo - found should be find
changed

Line 28: What do you mean with vanished? I would use another word here or rephrase it.

rephrased

Line 42: replace by with with.

done

Line 52: replace run with go

done

Line 53: replace the entire with all

done

Line 57-58: What do you mean with "divergent entities intervene"?

We rewrote to: "...or when genetically divergent sources interbreed"

Line 65: replace holding with comprising

corrected

Line 70-71: Rephrase. I would suggest: In this regard, the system offers an interesting setting as the possible source for colonizing the island are birds from ..."

done

Line 76: This island population is the most isolated population of Eurasian blackbirds in Central Europe. Do you have a reference for that?

No, yet since Heligoland is the most isolated island along the German coast (and there are no other islands in Poland) and given the continuous distribution of the blackbird on the mainland, we deem it not necessary to back to this statement with references.

Line 84: Mention that you used microsatellites here.

bracketed

Line 103-105: Move this sentence to the beginning of the section. It's important to know how many markers you used from the beginning.

We moved the sentence right after the first sentence introducing the amount of data.

Line 104: typo - polymorphic

changed

Line 143: It is not clear from the text how you performed the population assignment test. I guess you compared the STRUCTURE output with the origin of the samples.

We explicitly mentioned the assignment test in GenAlEx in LL119/120. To make this more clear, we added the program name in the sentence.

Line 175-177: Can you really distinguish between recurrent migration and several propagules that intermixed later on?

Yes. This sentence mentions the INITIAL colonization that was initiated by a single breeding pair (see previous sentence). Hence, a large number of putative propagules would have led to a higher number of breeding pairs in the initial colonization of the island, which is not supported by the data documenting the population development of the Heligolandian blackbirds (see refs 18,32 in the manuscript).

Line 186: replace was with were

changed

Line 187: I would not use the word "genotype" here. Cluster would be a better term (see also Line 204 and 206).

changed

Line 188-190: How do your measures of genetic divergence relate to other studies on island populations? It would be interesting to compare your F_{st} -values with previous studies.

We totally agree that this would be very interesting. However we think that such a comparison would go beyond the scope of our paper. This is mainly because island-mainland differentiation in other avian systems might be affected by a lot of different factors such as: 1) island size and 2) distance to the mainland, 3) general mobility of the species (i.e. migratory, sedentary), 4) functional resistance of water bodies as perceived by the species, 5) time since colonization of the island, 6) functional connectivity (i.e. effective migration), etc. Such a comparison could be subject of a (very interesting) review. Hence, in order to not selectively pick studies that confirm our findings - which would generate a skewed impression - we prefer to not include more studies apart from our species-specific expectations. Yet, to clarify the used references, we mentioned our study species here.

Line 195: replace elaborated with identified

changed

Line 212-213: Because you are dealing with one species, admixture might be a better term than hybridization (which implies different species or subspecies).

We agree and changed 'introgressive hybridization' to 'admixture'

Line 216-219: This sentence is difficult to follow. Please rephrase.

We clarified this sentence by adding a more concise explanation and splitting the sentence into two.

Literature:

Peery et al. 2012: Mol Ecol 21: 3403-3418.

Hoban et al. 2013: Mol Ecol 22: 3444-3450.

Reviewer: 2

Comments to the Author(s)

This is a very clear paper. I would only like to see more named examples elaborated on in Discussion (and Introduction) apart from one Zosterops and Darwin's finches.

We would like to thank the reviewer for the constructive feedback.

Regarding the general comment of using more examples, we already have included examples from barn swallow, melodious warbler, purple martin, house finch and forest thrush in the introduction and discussion.

Mentioning all the species explicitly in the text would introduce too much distracting, review-like, information which is not of immediate relevance to the manuscript or to support any of the findings. In saying that, we revised the introduction and discussion in that we did not explicitly mention other species anymore while references remain. We found that this revision streamlines the manuscript and improved the reading.

minor issues include

23 clusters

changed

27 did not find
changed

28 (and more often): deviations from the Hardy-Weinberg equilibrium
Corrected

32 Which phenotypic traits namely? Song?
Since this is meant as an outlook rather than a concrete result from this manuscript we prefer to not mention explicit traits here in the abstract. Nevertheless, we know from available data from this population that many island birds have very round wings and variation in song parameters. This variation poses interesting opportunities to relate these data with our genetic findings. Hence, we add some more details in the final paragraph of the manuscript where we seize the idea again and cite a study on morphological and song variation in the island population.

48 What do you mean by "initial state"?
We changed 'state' by 'properties'.

52 Such genetic bottleneck is also called founder effect. Please add this term.
added

64 Heligoland is an archipelago, not a single island. Specify at least here, if blackbirds settled/were sampled on the main island only.
We specified the main island now in the following sentence and also add details on the blackbird population on the neighboring dune island.

109 (and more often in text and tables): Make sure to use subscripts and appropriate case in population genetic parameters.
changed

123 from one to ten
changed

139 deviation
Since we made multiple comparisons with HWE instead of just one, we think the plural term applies here and is kept as is.

Appendix B

Dear Editor,

Thanks once more for the neat review process. With a vacation-related delay I have now incorporated the final minor edits into the manuscript. Since there is no major concern we simply agreed with everything the reviewer suggested including a thorough check of the tense in the results section. On behalf of all coauthors I'm much looking forward receiving the proofs of the manuscript.

Kind regards,

Jan Engler et al.

Reviewer: 1

Comments to the Author(s)

The authors have nicely addressed all my concerns. They performed extra analyses to show that the deviations from Hardy-Weinberg equilibrium did not affect the STRUCTURE analyses. And they added a statistical procedure to the simulations. Although I would have liked a statistical test of a potential bottleneck, I understand that this was not feasible with the present data set. I think this manuscript is almost ready for publication. I did, however, find a few minor mistakes in the text (see below). Notably, the results section switches between past and present tense. This can easily be corrected.

Dear Reviewer,

We would like to thank you for your fair and constructive evaluation. We accounted for all remaining issues, so there will be no line-specific responses necessary.

Sincerely,

Jan Engler et al.

Minor comments

Line 76: remove from
Line 109: Individuals should be lower-case
Line 138: Structure should be STRUCTURE
Line 139: replace run with ran
Line 163: replace where with were
Line 219: replace of with into
Line 243: replace united with merged
Line 243: replace need with needs